# Raman Analysis of E₂ (High) and A₁ (LO) Phonon to the Stress-Free GaN Grown on Sputtered AlN/Graphene Buffer Layer

**Yu Zeng** [1,2]**, Jing Ning** [1,2,*]**, Jincheng Zhang** [1,2]**, Yanqing Jia** [1,2] **, Chaochao Yan** [1,2]**, Boyu Wang** [1,2] **and Dong Wang** [2]

[1] Department of the State Key Discipline Laboratory of Wide Band Gap Semiconductor Technology, University of Xidian in Xi'an, Taibai South Road, Xi'an 710071, China; yzeng2@stu.xidian.edu.cn (Y.Z.); jchzhang@xidian.edu.cn (J.Z.); yqjia@stu.xidian.edu.cn (Y.J.); ccyan@stu.xidian.edu.cn (C.Y.); wangbyxd@stu.xidian.edu.cn (B.W.)

[2] Shanxi Joint Key Laboratory of Graphene, School of Microelectronics, University of Xidian in Xi'an, Taibai South Road, Xi'an 710071, China; chankwang@xidian.edu.cn

* Correspondence: ningj@xidian.edu.cn

**Abstract:** The realization of high-speed and high-power gallium nitride (GaN)-based devices using high-quality GaN/Aluminum nitride (AlN) materials has become a hot topic. Raman spectroscopy has proven to be very useful in analyzing the characteristics of wide band gap materials, which reveals the information interaction of sample and phonon dynamics. Four GaN samples grown on different types of buffer layers were fabricated and the influence of graphene and sputtered AlN on GaN epitaxial layers were analyzed through the E₂ (high) and A₁ (LO) phonon. The relationship between the frequency shift of E₂ (high) phonons and the biaxial stress indicated that the GaN grown on the graphene/sputtered AlN buffer layer was stress-free. Furthermore, the phonon lifetimes of A₁ (LO) mode in GaN grown on graphene/sputtered AlN buffer layer suggested that carrier migration of GaN received minimal interference. Finally, the Raman spectra of graphene with the sputtered AlN interlayer has more disorder and the monolayer graphene was also more conducive to nucleation of GaN films. These results will have significant impact on the heteroepitaxy of high-quality thin GaN films embedded with a graphene/sputtered AlN buffer, and will facilitate the preparation of high-speed GaN-based optoelectronic devices.

**Keywords:** E₂ (high) phonon; A₁ (LO) phonon; stress-free GaN; graphene; sputtered AlN

## 1. Introduction

As a group-III nitrides semiconductor, gallium nitride (GaN) materials with superior properties such as wide electronic energy band, wurtzite structural symmetry, strong atomic bonds, high thermal conductivity and so on, have shown major potential in the manufacturing field of optoelectronic devices [1,2]. In recent decades, GaN has demonstrated broad development prospects in the application of high-brightness blue/green light emitting diodes (LEDs), blue laser diodes (LDs), photoelectric detectors and deep ultraviolet optoelectronic devices [3–5]. Although remarkable progress in the growth of GaN and the fabrication technologies of the GaN-based devices has been made, some basic properties of GaN such as phonon dynamics and interactions (phonon–electron and phonon–phonon) still need to be studied further. It is known that the the stress in the GaN epilayer caused by the lattice constants and thermal expansion coefficient mismatch between the substrate and GaN films is still an extremely critical issue in the current study [6–8]. The existence of stress will induce defects, which are considered as the primary factor restraining further development of high efficiency III-nitride

optoelectronic devices. In addition, the mechanism of lattice vibration and phonon scattering inside GaN is also crucial for the performance improvement of high-speed and high-power GaN-based devices. Raman spectroscopy is a non-contact, damage-free characterization method that does not require preparation of the sample in advance compared with other characterization methods. More importantly, the stress and the interaction between phonon and carrier properties of GaN can be characterized by Raman spectra without damage to the sample. For this reason, it is greatly helpful to explore the internal information of GaN crystals with Raman spectra. In addition, being proficient in the knowledge of phonon dynamics and interactions (electron–phonon or phonon–phonon) with free carriers of GaN is also an essential foundation to the principle of lattice scattering and the molecular vibration.

To date, in most of the Raman scattering studies performed for GaN, there have been research results showing that all symmetry-allowed optical phonons of the Raman spectra were detected in the sapphire substrate according to the polarization selection rules by T. Azuhata [9]. Furthermore, a high-pressure structural phase transition was observed in GaN at 47 GPa by means of Raman spectroscopy [10]. The coupled modes of the longitudinal optical (LO) phonon and overdamped plasma in the GaN grown with aluminum nitride (AlN) buffer layer by Metal-organic Vapor Phase Epitaxy (MOVPE) have also been studied in sapphire substrate [11], which indicated the manner in which the different carrier concentrations changed with the $A_1$ (LO) scattering peak shifts. We already know that depositing different buffer layers on the single crystal sapphire substrate will greatly affect the quality of the GaN epitaxial layer above it, which further affects the phonon scattering process in the GaN epitaxial layer. By studying the Raman phonon behavior in GaN films, it can provide a support for subsequent analysis and improvement about the quality of GaN films. In recent years, the use of graphene as a buffer layer to grow GaN epitaxial layers has attracted significant interest. Graphene possesses many properties that can improve the quality of GaN epitaxial layers and GaN devices [6–8,12–14]. However, there is no systematic theoretical study on the phonon scattering behavior in the GaN epilayer obtained by inserting graphene buffer layers.

In this work, we have used Raman spectroscopy to investigate the $A_1$ (LO) and $E_2$ (high) phonons in high-quality GaN thin films using different AlN buffer layers with the metal-organic chemical vapor deposition (MOCVD) two-step method. In the growth experiment of the AlN buffer layer, the AlN buffer layer was grown with both growth methods: the low temperature–high temperature MOCVD method and the sputtered method, respectively. Furthermore, the graphene layer was grown on the copper foil films using the chemical vapor deposition (CVD) method and then was wet-transferred from the copper foil films onto the different samples [15]. Analysis of relationship between the $E_2$ (high) phonon and strain provided strong evidence of the GaN epilayer inserted with graphene/sputtered AlN buffer was stress-free. Furthermore, the detailed discussion of the full-widths at half-maximum (FWHM) and $A_1$ (LO) phonon lifetimes showed that the carrier mobility in the GaN with graphene/sputtered AlN buffer layer shown less interference. In addition, comparing the D band, D+D″ band and 2D band that appeared in the Raman spectra of graphene transferred on the MOCVD AlN and sputtered AlN buffer layer, respectively, we concluded that monolayer graphene with more defects on the sputtered AlN buffer layer contributed more to the growth of the GaN thin films.

## 2. Materials and Methods

In this article, to conduct a systematic research study on the different buffer layers containing graphene layers, four different buffer layers were prepared between the GaN epilayer and sapphire substrate. The growth of GaN epilayers in all samples were prepared by using a conventional low temperature–high temperature two-step growth method of metal organic chemical vapor deposition (MOCVD). The buffer layer structure used in the four samples is shown in Table 1 and Figure 1 below. For the experiment preparation of sputtered AlN film, the growth temperature of the sputtered AlN buffer layer was set at 550 °C and the radio frequency (RF) sputter system pressure was adjusted to $10^{-7}$ Torr. Then, high-purity (99.99%) Al target was used to deposit 25-nm-thick sputtered AlN films in the radio frequency (RF) sputter system with 50 W of RF power. In the traditional two-step growth

method of MOCVD, the formation mechanism of the epitaxial layer is to nucleate at low temperature and coalesce into a thin film at high temperature. Therefore, the MOCVD AlN epitaxial layer was first nucleated at a lower temperature of 550 °C while the temperature increased to 1220 °C, coalescing into a film. Hexagonal honeycomb graphene was grown on the polished copper foils via chemical vapor deposition (CVD) and transferred on the surface of the sample via wet transfer. In the process of growing GaN films, trimethylgallium (TMGa) and ammonium ($NH_3$) were used as the source of Ga and N for the growth of GaN epilayer, respectively. Firstly, low temperature GaN films were deposited at about 550 °C and kept for 5 min. Then, the temperature was raised to 1050 °C and heated for 70 min to form high temperature GaN films, and finally a GaN epitaxial layer with a thickness of about 2 μm was obtained.

**Table 1.** Schematic table of buffer layer structure information of four samples in this work.

| Number | Substrate | Buffer Layer | Epitaxial Layer |
|--------|-----------|--------------|-----------------|
| sample 1 | sapphire | sputtered aluminum nitride (AlN)/graphene | gallium nitride (GaN) |
| sample 2 | sapphire | MOCVD AlN/graphene | GaN |
| sample 3 | sapphire | graphene/MOCVD AlN | GaN |
| sample 4 | sapphire | MOCVD AlN | GaN |

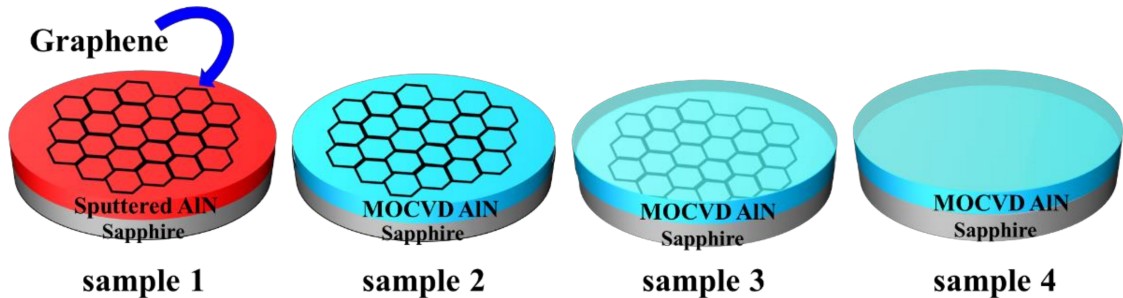

**Figure 1.** Schematic diagram of the sapphire substrate structure with different buffers: sample 1 with graphene/sputtered AlN buffer, sample 2 with graphene/MOCVD AlN buffer, sample 3 with MOCVD AlN/graphene buffer, and sample 4 with MOCVD buffer.

Experimental analysis provided reliable evidence that the stress in GaN could be indicated by the $E_2$ (high) phonon peak in the Raman spectrum. Furthermore, the full width at half maximum (FWHM) of the Raman scattering peak of $A_1$ (LO) also better characterized the process of Raman scattering in GaN from the perspective of phonon lifetime. On the one hand, the frequency shift of the $E_2$ (high) phonon was analyzed, so that we could deduce the magnitude of the strain in the GaN films grown with different buffer layers. If the $E_2$ (high) phonon peak moved to the lower frequency relative to the corresponding peak position of stress-free GaN films, it indicated that there was tensile stress in the GaN films. Otherwise, we judged that the stress that GaN films subjected to was compressive stress. [16,17]. On the other hand, the destruction of the long-range order due to the defects or impurities would impair the intensity of $E_2$ (high) phonon peak and $A_1$ (LO) phonon peak. The above phenomenon would also be manifested as the broadening of the phonon peak. Therefore, through the energy-time uncertainty relation, the phonon lifetime was determined by observing the phonon line width of the $A_1$ (LO) phonon [18,19]. In addition, the Raman spectra of the GaN layer was measured at room temperature in the range from 300 cm$^{-1}$ to 800 cm$^{-1}$ using micro-Raman testing technology with back scattering geometry. We focused the 514 nm line of an Ar+ laser onto a 10-um-diameter spot and used it as the excitation source, and the scattered light was detected by the charge-coupled detector (CCD). The Raman spectra resolution was better than 0.1 cm$^{-1}$ to ensure the accuracy of the measured spectrum.

### 3. Results

In the initial stage of GaN epitaxial layer growth, the hydrogen in the reaction chamber would etch the graphene film on the substrate surface, which increased the defects of the surface for graphene and promoted the nucleation of the GaN film. Therefore, we could analyze the information between the characteristic peaks of graphene in the Raman spectrum to reflect the defects of graphene. The Raman intensity ratio of the 2D/G mode indicated that the graphene transferred on the sputtered AlN and MOCVD AlN layer was the monolayer, which showed that the single layer graphene was stable. Because graphene exhibited similar in-plane lattice arrangements as wurtzite structure GaN, the single-layer graphene was more conducive to remote epitaxy and guided the nucleation of the GaN epilayer. Meanwhile, weak van der Waals forces on both sides of the graphene shielded the stress resulting from lattice mismatch and thermal expansion coefficient mismatch between the GaN epilayer and sapphire substrate, thus improving the crystalline quality of the GaN epilayer. Through analyzing the $E_2$ (high) and $A_1$ (LO) phonons in GaN epitaxial layer grown by MOCVD with four different types of buffer layers, the peak shift of the $E_2$ (high) phonon peak was found to be the lightest shift, which suggested that the stress that the GaN films grown on the graphene/sputtered AlN buffer layer suffered was lowest. In addition, the phonon lifetimes reflected through the $A_1$ (LO) phonon peak were shortest when GaN film was directly deposited on the graphene/sputtered AlN buffer layer. In this case, it is beneficial to the carrier migration in the GaN with the graphene/sputtered AlN buffer layer. Through Raman analysis of GaN and graphene, it is concluded that the GaN obtained on the graphene/sputtered AlN buffer layer was subject to stress-free conditions, and the single-layer graphene obtained by the wet-transfer was also more conducive to nucleation in the growing process. This work will play an important role in the subsequent manufacture of GaN-based optoelectronic devices dominated by optical-phonon and carrier interaction.

#### 3.1. Raman Spectrum of Graphene

Figure 2 shows a typical Raman spectrum of graphene transferred with the sputtered AlN layer or MOCVD AlN interlayer on the sapphire substrate. Under the laser light, the electrons in the graphene jumped from the valence band to the conduction band, and the electrons interacted with the phonons to scatter, resulting in different Raman characteristic peaks. In both growth conditions, the peaks at 1350, 1584 and near 2700 $cm^{-1}$ were characteristic peaks related to the D, G, and 2D modes of graphene, respectively. The observation of the so-called G band, which was the graphene Raman active mode related to the stretching of the C–C bonds, appearing at 1584 $cm^{-1}$, was also important. In addition, the existence of an obvious and sharp G scattering peak indicated that the two-dimensional graphene layer was well present in the two structures. According to the principle of Raman phonon scattering of graphene, the G scattering peak (1584 $cm^{-1}$) was generated by the in-plane vibration of the $SP_2$ carbon atom and the interaction between the dual degenerate in-plane transverse optical (iTO) and in-plane longitudinal optical (iLO) optical phonons in the center of the Brillouin zone [20,21]. In addition, pure and single graphene layer did not contain disordered states, and the Raman spectra contained only G mode, D + D″ mode and 2D mode [22]. If the lattice size decreased, the long-range periodicity of the graphene was destroyed, proving that there appeared some defects and disorders in the graphene layer. Therefore, the D mode would be induced as a result of the disorders and defects existing in the graphene. Furthermore, the greater the degree of disorder, the higher the intensity of the D mode as well as the ratio of $I_D/I_G$. In addition, the type of defects could also be judged by the ratio of the intensity of the $I_D/I_{D'}$ mode. (If the density of defects is very high, the D′ mode will appear at 1620 $cm^{-1}$). However, the ratio of $I_D/I_G$ would decrease if the carbon structure was more amorphous, which led to the attenuation of all Raman peaks [23,24]. Moreover, defects were not only characterized by the appearance of new defect-induced peaks but also the broadening of the peak in the Raman spectrum. As shown in Figure 2, the D band appeared in the both growing conditions, indicating that the disorders emerged in graphene in both growing conditions. Furthermore, using the basic principle of Raman spectroscopy to characterize the defects caused by the destroyed long-range

periodicity in a two-dimensional (2D) system, we mainly used the intensity of D + D″ mode to estimate the density of the defects. Therefore, from the Raman spectrum, the intensity of D + D″ scattering peak was stronger as graphene on the sputtered AlN layer, indicating that the high density of defects on the graphene/sputtered AlN buffer was more beneficial for nucleation during the growth of GaN. Additionally, according to experiment method, before the growth process of GaN films, the N source ($NH_3$) etched the graphene surface and introduced defects and nucleation sites for subsequent epitaxial layer growth, so the intensity of the defect-induced D scattering peak was not very low. Besides, it is acknowledged that the intensity of 2D in the Raman spectra of graphene reflected the number of graphene layers. Thus, we came to the conclusion that the graphene transferred on the sputtered AlN interlayer was a single layer due to the intensity of 2D being large and sharp, and that when the GaN epilayer was grown on graphene/MOCVD AlN/sapphire composite substrate, the wet-transferred graphene was a not a monolayer layer due to the wrinkles caused by the wet-transferred process of graphene. It is known that single graphene and III nitrides were combined with each other by van der Waals forces. More importantly, graphene exhibited similar in-plane lattice arrangements as wurtzite structure GaN, so the single-layer graphene was more conducive to remote epitaxy and guided the nucleation of the GaN epilayer. Meanwhile, weak van der Waals forces on both sides of the graphene shielded the stress resulting from lattice mismatch and thermal expansion coefficient mismatch between the GaN epilayer and sapphire substrate, thus improving the crystalline quality of the GaN epilayer. In summary, the interaction between the sputtered AlN layer and graphene was more beneficial to the two-step growth of the high-quality GaN epitaxial layer above it.

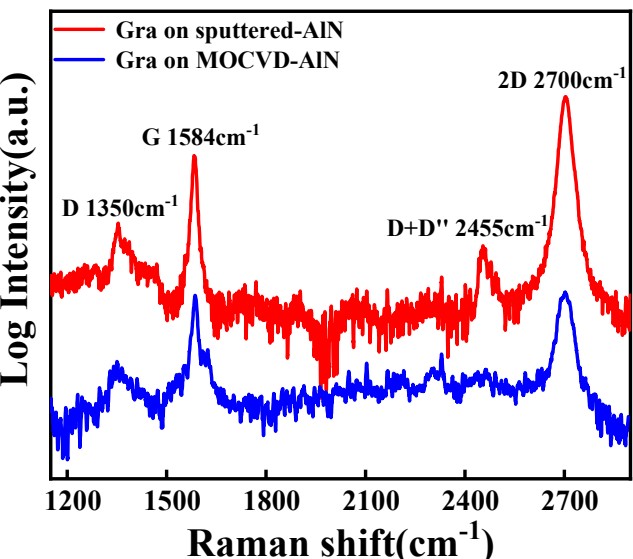

**Figure 2.** Raman spectra of graphene transferred on the sputtered AlN layer (red line) and MOCVD AlN (blue line) layer in the high-wavenumber region of 1200–2800 $cm^{-1}$. The D, G, and 2D peak positions of graphene are indicated.

### 3.2. Raman Spectrum of GaN Epilayer

From Figure 3, it is clearly indicated that the scattering peak at 417 $cm^{-1}$ represents the typical characteristics of sapphire ($Al_2O_3$) and the five characteristic phonon scattering peaks symbolize the structure of wurtzite crystal GaN with the different buffer layers, which were $A_1$ (TO) phonon at 532 $cm^{-1}$, $E_1$ (TO) phonon at 557 $cm^{-1}$, $E_2$ (high) phonon at 568.2 $cm^{-1}$, $A_1$ (LO) phonon at 736 $cm^{-1}$ and $E_1$ (LO) phonon at 750 $cm^{-1}$ in the range between 300 and 800 $cm^{-1}$, respectively [25–27]. According to the selection rule, only the $A_1$ (LO) phonon and $E_2$ (high) phonon should appear in the Raman spectrum of c-plane GaN, but the crystal grains might be twisted during the epitaxy process of GaN, which caused lattice deformation or structural disorders to some extent, resulting in the loosening

of the selection rule and the appearance of phonon modes that should be prohibited. Thus, both the Raman spectrum of samples 1 and 2 displayed a low intensity peak between 500 cm$^{-1}$ and 550 cm$^{-1}$ along with a shoulder in the E$_2$ (high) mode in Figure 3. We thought that the appearance of the A$_1$ (TO) phonon (532 cm$^{-1}$) and E$_1$ (TO) phonon (557 cm$^{-1}$) was interesting. We found clearly that the intensity of the E$_2$ (high) phonon became increasingly strong and that there was a shift to lower frequency with the introduction of the graphene layer between the GaN epitaxial layer and the sapphire substrate, which compared well with previous results [3], revealing that the quality of the GaN epilayer had been greatly improved with the help of graphene. Furthermore, it is also meaningful for us to investigate the phonon lifetimes of the A$_1$ (LO) phonon based on the principle of wave vector conservation using the A$_1$ (LO) phonon line widths [28]. Studying the relationship between phonon lifetime and FWHM of peaks was very important to characterize the degree of disorder within the GaN, and it was beneficial to use the Raman spectrum to characterize the properties of GaN materials in our subsequent research.

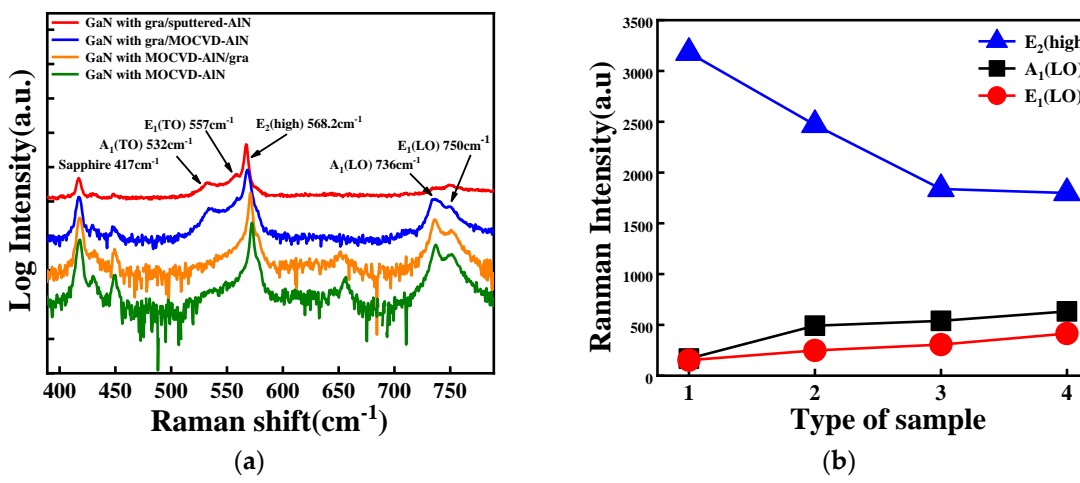

(**a**)　　　　　　　　　　　　　　　(**b**)

**Figure 3.** (**a**) Raman spectra of the GaN films grown on sapphire substrate with graphene/sputtered AlN buffer layer (sample1), graphene/MOCVD AlN buffer layer (sample 2), MOCVD AlN/graphene buffer layer (sample 3), and MOCVD AlN buffer layer (sample 4) in the low-wave region of 300–800 cm$^{-1}$; (**b**) the intensity of the A$_1$ (LO) phonon, E$_1$(LO), and E$_2$ (high) of Raman spectra of the GaN films in the four different samples.

### 3.3. Raman E$_2$ (High) Phonon of GaN Epilayer

For the E$_2$ vibration mode, the positive and negative charges in the unit cell are not shifted, so the E$_2$ vibration mode was a non-polar mode. Moreover, since the E$_2$ phonon mode corresponded to the stretching and compression of the atoms in the lattice, so the E$_2$ (high) phonon could characterize the biaxial strain existing in the GaN [17,29,30]. Generally, the large lattice and thermal mismatches between these substrates and epitaxial layer lead to a high defect density and large biaxial stress, and the residual stress will affect the phonon scattering frequency. Furthermore, the frequency shift of the E$_2$ (high) phonon was very sensitive to the biaxial strain in GaN layer, and the stress in GaN therefore could be measured based on the Raman shift of the E$_2$ (high) scattering peak in the geometry scattering configuration (z(y,y)z). Figure 4a clearly shows the E$_2$ (high) phonon scattering peak's position in the spectrum of the GaN film grown on four different types of substrates. From the Figure 4a, we found that when the GaN epilayer was grown on the graphene/sputtered AlN/sapphire substrates, the frequency of E$_2$ (high) phonon was 568.2 cm$^{-1}$, close to the 568 cm$^{-1}$ corresponding to the stress-free GaN films [31]. Therefore, we thought the GaN epilayer with graphene/sputtered AlN buffer layer was almost stress free. However, when the GaN epilayer was grown on sapphire substrates without sputtered AlN buffer layer (sample 2–4), the frequency of the E$_2$(high) phonon exhibited movement toward the high frequency (569.1 cm$^{-1}$, 570.7 cm$^{-1}$, 572.3 cm$^{-1}$, respectively), compared with stress-free GaN films (568.0 cm$^{-1}$) [31], showing that the GaN epilayer was subjected

to compressive stress. On the contrary, the degree of frequency shift of the $E_2$ (high) phonon scattering peak in the GaN films with graphene/sputtered AlN buffer layer was so slight that it was almost close to the frequency value of stress-free (568 cm$^{-1}$) [31], indicating evidently that the stress induced by defects and lattice mismatch in GaN films was lowest under this growth mode. Moreover, the release of stress due to graphene introduction could be estimated to a large extent by the following formula,

$$\Delta\omega = K\sigma, \tag{1}$$

in which $\Delta\omega$ is the difference of the $E_2$ (high) peak with the stress-free GaN crystal value of 568.0 cm$^{-1}$ [31], K represents the stress coefficient $\approx$2.56 cm$^{-1}$ GPa$^{-1}$, and $\sigma$ is the stress value of the GaN epitaxial layer [32]. It is acknowleged that the biaxial stress value of GaN films grown on the sapphire substrate with graphene/sputtered AlN buffer layer (sample1) was 0.07 GPa and was clearly the lowest compared to the other samples (2–4) in Figure 4d, suggesting that the effective stain relaxation in the GaN epitaxial layer by the introduction of graphene films and the sputtered method would create more nucleation points than the MOCVD method.

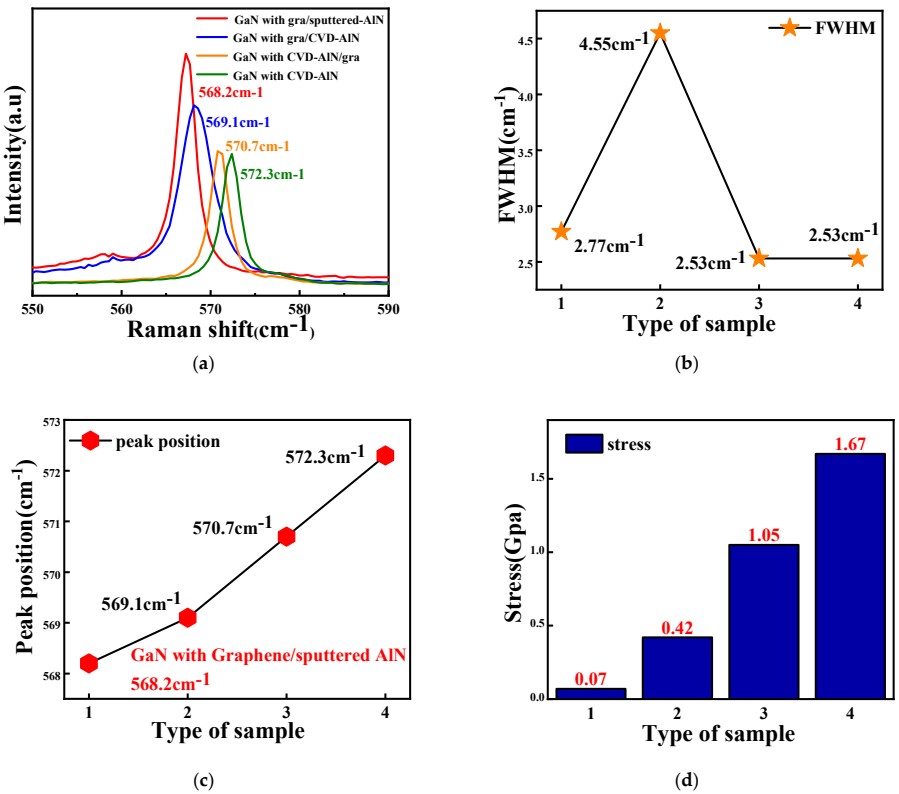

**Figure 4.** (**a**) $E_2$ (high) phonon of Raman spectra of GaN films grown on sapphire substrate with graphene/sputtered AlN buffer layer (sample 1), graphene/MOCVD AlN buffer layer (sample 2), MOCVD AlN/graphene buffer layer (sample 3), and MOCVD AlN buffer layer (sample 4) in the low-wave region of 550–590 cm$^{-1}$; (**b**) full-widths at half-maximum (FWHM) and (**c**) peak position of $E_2$ (high) peak of Raman spectra of the GaN films in the four different samples; (**d**) the stress value of GaN films in the four different samples.

A large number of studies have shown that the phonon intensity and the full width at half maximum (FWHM) of the $E_2$ (high) phonon scattering peak reflected the discrepancy in the crystal quality of the material. The GaN films with high dislocation density had smaller strength and larger full width at half maximum (FWHM). It could be seen from the Figure 4a,b that the FWHM of the $E_2$ (high) phonon scattering peak was distinctly the narrowest and the intensity was largest on the substrate covered by the graphene/sputtered AlN buffer layer; the reason for the above phenomenon is the

differential thermal expansion coefficient between the sapphire ($Al_2O_3$) substrate and the AlN buffer leading to the stress in the GaN films; furthermore, the lower the growth temperature (550 °C) in the sputtered AlN buffer, the smaller effect of sapphire substrate on the $E_2$ (high) phonons. The graphene films also facilitated the formation of uniform GaN films. It is demonstrated that the graphene/sputtered AlN buffer layer effectively improved the quality of the GaN thin film and promoted less scattering of $E_2$ (high) phonons by dislocations and defects.

### 3.4. Raman $A_1$ (LO) Phonon of GaN Epilayer

As shown in Figure 5, observing the line width of the $A_1$ (LO) phonons of Raman spectra of GaN with different interlayers, the $A_1$ (LO) phonon lifetime can be determined from the FWHM based on the energy–time uncertainty relationship [33,34], in order to analyze the growth factors that affected the GaN Raman scattering. The formula about FWHM and phonon lifetime is as follows [34].

$$\frac{\Gamma}{\hbar} = \frac{1}{\tau}, \tag{2}$$

in which $\tau$ was the phonon lifetimes and unit was picoseconds, $\hbar$ was the modified Planck constant ($5.3 \times 10^{-12}$ $cm^{-1}$ s), and $\Gamma$ was representative to the true FWHM in units of $cm^{-1}$. With graphene/sputtered AlN interlayer, it is noted that $A_1$ (LO) phonons in the Raman spectrum (Figure 5a) were almost stretched, so the FWHM of the $A_1$ (LO) phonon scattering peak was almost invisible in this case. In addition, the FWHM of $A_1$ (LO) phonon mode was larger as the quality of the buffer layer improved. Under the constraint of the law of conservation of energy, Figure 5b was drawn by using the relationship between the FWHM of $A_1$ (LO) phonon and phonon lifetimes [34]. It was very clear that $A_1$ (LO) phonon lifetimes were shorter after the addition of the graphene layer, and that $A_1$ (LO) phonon lifetimes were shortest when GaN films were directly deposited on the graphene/sputtered AlN buffer layer (sample 1), which was in good agreement with the above results of Figure 5a. A more critical issue was the sufficient lifetime of longitudinal optical (LO) phonons greatly weakening the rate at which hot electrons released energy and inhibited carrier mobility. Hence, the $A_1$ (LO) phonon mode of Raman spectrum of GaN grown with graphene/sputtered AlN buffer layer had the shortest phonon lifetimes, which were most conducive to carrier migration in this situation. This also reflected that during the thermal movement of carriers, the probability of collisions with crystal lattices, impurities, defects, etc. was reduced, and that the probability of scattering was also reduced. The lifetime variations of $A_1$ (LO) phonons of GaN had been explored on different buffer layers, which contributed more to the subsequent research and preparation of high-speed optoelectronic devices dominated by photon–phonon carrier interaction on GaN films.

### 3.5. X-ray Diffraction of GaN Epilayer

The density of dislocations of GaN epilayers grown on graphene/sputtered AlN/sapphire and graphene/MOCVD AlN/sapphire substrates were further evaluated via X-ray diffraction (XRD). It is well known that dislocations in the materials caused the distortion to crystallographic planes and were accompanied by the broadening of the FWHM in the corresponding XRD $\omega$-scan (rocking curves). The screw and edge dislocations could be determined by obtaining the FWHM in the XRD $\omega$-scan (rocking curve). Figure 6a,b showed the (0002) and (10–12) reflection peaks of GaN films grown on graphene/sputtered AlN/sapphire and graphene/MOCVD AlN/sapphire substrates, respectively. The XRD rocking curves demonstrated that the FWHMs in both (0002) and (10–12) reflection peaks were narrower for the GaN films with the graphene/sputtered AlN/ buffer layer than those with the graphene/MOCVD AlN buffer layer, indicating that the densities of screw and edge dislocations in the GaN film grown on graphene/sputtered AlN/sapphire substrate had been reduced. Generally speaking,

the densities of dislocations in the XRD $\omega$-scan (rocking curve) for GaN films could be estimated using the following empirical formula [35].

$$N = \frac{\beta^2}{4.35 \times |b^2|}, \tag{3}$$

in which $\beta$ was the FWHM of the XRD $\omega$-scan (rocking curve) and $b$ was the Burgers vector of the corresponding dislocations. The (0002) FWHM of the X-ray x-scan (rocking curve) of the GaN epilayer was significantly reduced from 542 to 133 arcsec with the assistance of the graphene/sputtered AlN buffer layer, and the (10–12) FWHM was reduced from 597 to 246 arcsec. The estimated densities of the screw and edge dislocations of the GaN epilayer with the graphene/MOCVD AlN buffer layer were $2.85 \times 10^8$ and $3.45 \times 10^8$ cm$^{-2}$, respectively. In addition, the reduction in dislocation had been to $1.71 \times 10^5$ and $5.87 \times 10^7$ cm$^{-2}$ with the assistance of the graphene/sputtered AlN buffer layer. Therefore, the stress of the epitaxial layer was largely released, and the decreasing of dislocation density in the GaN epilayer could be attributed to the alleviation of the mismatch effects between the substrate and GaN epilayer. Furthermore, in the growth of the sputtered AlN buffer, the collision between electrons and gas molecules was strong in the magnetic field, so that the growth rate was greatly improved, which was helpful to increase the energy of the atoms on the surface of the substrate and improve the quality of AlN film to a large extent compared to the MOCVD AlN film. It could be seen that the results of XRD were consistent with the Raman spectrum.

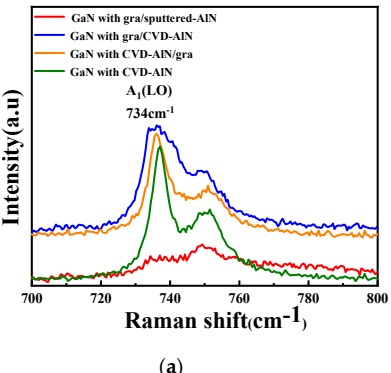 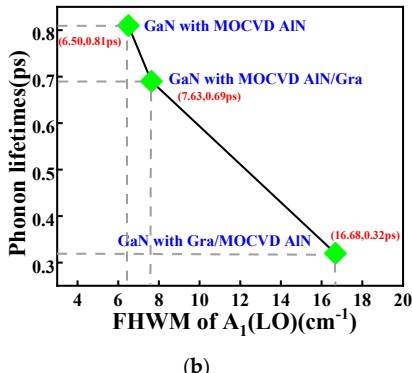

(**a**)　　　　　　　　　　(**b**)

**Figure 5.** (**a**) A$_1$ (LO) peak of Raman spectra of GaN films grown on sapphire substrate with graphene/sputtered AlN buffer layer (sample 1), graphene/MOCVD AlN buffer layer (sample 2), MOCVD AlN/graphene buffer layer (sample 3), and MOCVD AlN buffer layer (sample 4) in the wave region of 700–800 cm$^{-1}$. (**b**) The relationship between the FWHM of A$_1$ (LO) phonon and the phonon lifetimes in the four different samples.

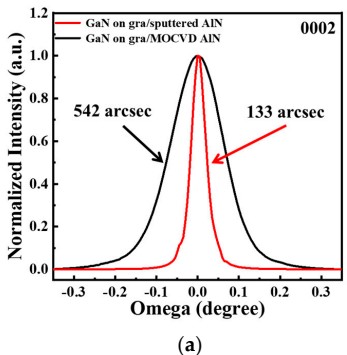 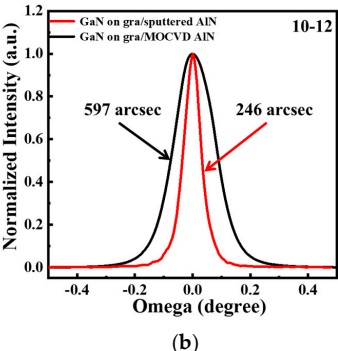

(**a**)　　　　　　　　　　(**b**)

**Figure 6.** (**a**) (002) and (**b**) (102) rocking curves of the GaN epilayer grown on graphene/sputtered AlN/sapphire and graphene/MOCVD AlN/sapphire substrates.

## 4. Conclusions

In conclusion, we analyzed the $E_2$ (high) and $A_1$ (LO) phonons of the GaN epitaxial layer grown on the single crystal sapphire substrate by MOCVD with four different types of buffer layers. We found that the peak shift of the $E_2$ (high) phonon peak was lightest, and the results of XRD indicated that the dislocations and defects in the GaN films grown on the graphene/sputtered AlN buffer layer had been reduced. The phonon lifetimes of the $A_1$ (LO) phonon peak were shortest when GaN films were directly deposited on the graphene/sputtered AlN buffer layer, which was most conducive to carrier migration in this situation. Finally, the Raman intensity of 2D mode indicated that the graphene transferred on the sputtered AlN buffer was monolayer, which was more helpful to remote epitaxy and guided the nucleation of the GaN epilayer. Through Raman analysis of GaN, it is concluded that the GaN obtained on the graphene/sputtered AlN buffer layer was subject to stress-free conditions, and the single-layer graphene obtained by the transfer was also more conducive to nucleation in the growing process. This work will play an important role in the subsequent manufacture of GaN-based optoelectronic devices dominated by optical-phonon and carrier interaction.

**Author Contributions:** Conceptualization, Y.Z. and J.N.; methodology, C.Y., J.Z., J.N. and D.W.; investigation, C.Y., B.W. and Y.J.; resources, J.Z., J.N. and D.W.; data curation, Y.J and Y.Z.; writing—original draft preparation, Y.Z.; writing—review and editing, J.N.; project administration, J.Z.; All authors have read and agreed to the published version of the manuscript.

**Funding:** This research was funded by the National Key Science & Technology Special Project (Program No. 2017ZX01001301), the Natural Science Basic Research Plan in Shaanxi Province of China (Program No. 2019ZDLGY16-08, 2019ZDLGY16-03, 2019ZDLGY16-02); Youth Science and Technology Nova Program of Shaanxi Province (Program No. 2020KJXX-068); the Wuhu and Xidian University special fund for industry-university-research cooperation (Program No. HX01201909039) and Fundamental Research Funds for the Central Universities (Program No. JBF201101).

**Conflicts of Interest:** The authors declare no conflict of interest.

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
