# Peer review of "Raman Analysis of E2 (High) and A1 (LO) Phonon to the Stress-Free GaN Grown on Sputtered AlN/Graphene Buffer Layer"

_applsci, doi:10.3390/app10248814_

Round 1

Reviewer 1 Report

The reviewed work deals with an important topic of stress analysis in heteroepitaxial layers. Despite many years of research and numerous works on nitride structures, new solutions for creating high-quality nitride electronic devices are still sought. The main idea of ​​the article fits perfectly into this research trend. The authors use graphene buffer layers to try to reduce the stress between the subsequent layers.
The topic of the use of graphene has already been presented by the authors, and analyzing the current works of the authors, it is difficult to say whether this research is a continuation or a minor supplement to the existing results. It seems that some data have already been published by authors. Nevertheless, I believe that the article is worth to be published. Of course, it needs to be improved and developed in many places.
In the attached file I am sending all comments to the article.

Reviewer 2 Report

The authors’ conclusion that GaN obtained on graphene/sputtered AlN buffer later was subject to stress free does play an important role in the manufacture of GaN based devices. However, I have several questions as following:

  1. Is there graphene on sample 4? If no, please correct the figure 1.
  2. Why does the wet transferred graphene exhibits monolayer only on sputtered AlN , and double layer on MOCVD grown AlN?
  3. To show the better crystalline quality of GaN grown on graphene/sputtered AlN, it is important to demonstrate the XRD evidence in addition to Raman. Please add it in the manuscript.

Reviewer 3 Report

  • Both samples 1 and 2 display a low intensity peak between 500 to 550 cm-1 along with a shoulder in the E2(high) mode in figure 3, which suggest some sort of lowering of symmetry due to possible existence of structural disorders.
  • The line shape and the blue shift of A1 (LO) peak can be related to stress as well as the free carrier concentration. However, the invisible A1 mode in sample 1 also suggests the changes in the electronic structure due to the presence of stacking disorders.
  • One can verify this by simple phase identification of all samples using the XRD technique. The presence of structural disorders in a given sample would cause non-uniform broadening of the XRD reflections.

Round 2

Reviewer 2 Report

Thanks for the revisions. 

The manuscript is much better.